# A Phase I Study of Carfilzomib with Cyclophosphamide and Etoposide in Relapsed and Refractory Leukemia and Solid Tumors

**DOI:** 10.3390/cancers17172924

**Published:** 2025-09-06

**Authors:** Jessica Boklan, Anne-Marie Langevin, Kevin Bielamowicz, Kathleen Neville, Tanya Trippett, Valerie Brown, Steven G. DuBois, Francis Eshun, Jonathan Gelfond, Ativ Zomet, Aru Narendran, Norman J. Lacayo

**Affiliations:** 1College of Medicine, University of Arizona, Phoenix, AZ 85004, USA; 2Health Science Center at San Antonio, University of Texas, San Antonio, TX 78229, USA; langevin@uthscsa.edu (A.-M.L.); gelfondjal@uthscsa.edu (J.G.); 3Arkansas Children’s Hospital, Little Rock, AR 72202, USA; kjbielamowicz2@uams.edu (K.B.); kathneville@gmail.com (K.N.); 4Memorial Sloan-Kettering Cancer Center, New York, NY 10065, USA; trippet1@mskcc.org; 5Penn State Hershey Children’s Hospital, Hershey, PA 17033, USA; vbrown1@hmc.psu.edu; 6Dana-Farber/Boston Children’s Cancer and Blood Disorder Center, Boston, MA 02115, USA; steven_dubois@dfci.harvard.edu; 7Phoenix Children’s Hospital, Phoenix, AZ 85016, USA; feshun@phoenixchildrens.com; 8Stanford Cancer Institute, Lucile Packard Children’s Hospital at Stanford, Stanford University, 750 Welch Rd., Ste. 224, Palo Alto, CA 94304, USA; azomet@stanford.edu; 9Alberta Children’s Hospital, Calgary, AB T3B 6A8, Canada; anarendr@ucalgary.ca

**Keywords:** pediatric phase 1, carfilzomib with cyclophosphamide/etoposide

## Abstract

A multicenter phase 1 study of carfilzomib administered in combination with cyclophosphamide and etoposide was conducted. The aim was to evaluate the safety of adding carfilzomib to chemotherapy commonly used in leukemia and solid tumors. We had pre-clinical evidence that this combination was active in cell lines from pediatric leukemias and solid tumors. We treated 38 leukemia and solid tumor patients between the ages of 6 and 17 years. A 5-day schedule of carfilzomib, cyclophosphamide, and etoposide was well-tolerated in patients with solid tumors, and at a lower dose of carfilzomib in leukemia. Furthermore, it was easily administered in an outpatient setting, making it a viable alternative to salvage therapy for children, adolescents, and young adults. This schedule warrants follow-up in patients with sarcomas as they benefited the most.

## 1. Introduction

Carfilzomib, a second-generation proteasome inhibitor, is a tetrapeptide ketoepoxide-based inhibitor that is specific to the chymotrypsin-like active site of the 20S proteasome. It is structurally and mechanistically distinct from the dipeptide boronic acid proteasome inhibitor bortezomib [1,2,3]. The proteasome is a protein-destroying apparatus involved in many essential cellular functions, such as regulation of the cell cycle, cell differentiation, signal transduction pathways, antigen processing for appropriate immune responses, stress signaling, inflammatory responses, and apoptosis. The proteasome is a central component of the protein degradation machinery in eukaryotic cells. Both transformed and normal cells depend on the function of the proteasome to control the expression of proteins linked to cell survival and proliferation.

In adults with multiple myeloma, a range of dosing strategies for carfilzomib have been used, and the most common adverse events include fatigue, neutropenia, nausea, diarrhea, thrombocytopenia, and dyspnea [4,5]. Carfilzomib has demonstrated significant in vitro activity alone and in combination with other antineoplastic agents against a panel of pediatric leukemic and solid tumor cell lines, including neuroblastoma, Ewing sarcoma, osteosarcoma, rhabdomyosarcoma, and atypical teratoid rhabdoid tumor (ATRT) [6,7,8,9,10].

In this study, carfilzomib was combined with a chemotherapy backbone of cyclophosphamide and etoposide, with all three agents administered daily for five days. The rationale for the use of this backbone includes the following: (1) in vitro studies demonstrated the synergy of carfilzomib in combination with etoposide and, synergy between cyclophosphamide and proteasome inhibitors [6]; (2) both cyclophosphamide and etoposide are active agents in lymphoid and myeloid pediatric leukemias and solid tumors [11,12]; (3) the combination of cyclophosphamide and etoposide has been used as a standard regimen to treat both solid tumors and leukemias [13,14]; (4) because of extensive experience with this chemotherapy backbone, the baseline toxicities are well-known [14]; (5) due to the rapid clearance in adult trials, it was thought that giving carfilzomib sequentially with the standard chemotherapy would maximize synergy [15]; and (6) since the cyclophosphamide and etoposide regimen is routinely administered daily for 5 days, it could be administered on the same days as carfilzomib, utilizing the 5-day dosing schedule of the initial phase 1 adult carfilzomib trial [16,17]. 

Based on adult carfilzomib trial results [4,15,18], a phase 1 clinical trial of carfilzomib administered in combination with cyclophosphamide and etoposide was undertaken to (1) determine the dose-limiting toxicities (DLTs) and the maximum tolerated dose (MTD) of carfilzomib administered in combination with cyclophosphamide and etoposide in pediatric patients with relapsed/refractory leukemia and solid tumors; (2) evaluate the toxicities of carfilzomib in the pediatric population when combined with conventional chemotherapy; and (3) gather preliminary efficacy data on the drug combination.

## 2. Methods

### 2.1. Eligibility Criteria

Patients aged 6 months to <30 years were eligible if they had relapsed/refractory leukemia, without central nervous system (CNS) 3 involvement, in second or greater relapse or who had failed at least one re-induction attempt after relapse or refractory disease (stratum A) or a relapsed/refractory non-CNS solid tumor including lymphoma (stratum B). Patients with CNS tumors were excluded due to lack of CNS penetration of carfilzomib [19]. For patients with solid tumors, measurable disease was not required. Other eligibility criteria included (1) a life expectancy ≥of 3 months; (2) a Karnofsky/Lansky performance of ≥50; (3) adequate bone marrow function (absolute neutrophil count ≥ 750/µL, platelets ≥ 75,000/µL, and hemoglobin ≥ 10 g/dL with or without transfusion, unless cytopenia was secondary to leukemia or bone marrow infiltration with solid tumor); (4) adequate liver function (AST and ALT ≤ 3× upper limit of normal (ULN) for age; total bilirubin ≤ 1.5 × ULN for age; direct bilirubin ≤ ULN for age unless elevation can be clearly attributed to liver leukemia or metastases); (5) adequate renal function (serum creatinine ≤ ULN for age or a glomerular filtration rate ≥ 70 mL/min/1.73 m^2^); (6) an echocardiography shortening fraction ≥ 27%; and (7) pulse oximetry measuring ≥ 95% saturation without supplemental oxygen. Specific eligibility and exclusion criteria are provided in detail in the POE14-01_Publication Appendix A. Informed consent was obtained from the patients or their legal guardians before study entry in accordance with individual institutional policies.

### 2.2. Treatment and Dose Escalation

Treatment consisted of five consecutive days of cyclophosphamide 440 mg/m^2^ IV over 60 min from hours 0 to 1, etoposide 100 mg/m^2^ IV over 120 min from hours 1 to 3, then carfilzomib IV over 30 min from hours 3 to 3.5. Granulocyte colony-stimulating factor (G-CSF) was started on day 6, 24–36 h after completion of the carfilzomib infusion. Cycles were at least 28 days long with no maximum number of cycles a patient could receive. For dose levels 4 and 5, patients were given carfilzomib at a dose of 20 mg/m^2^ on days 1–2 followed by an escalation to 27 mg/m^2^ or 36 mg/m^2^, respectively, for days 3–5. This ramp up to full dose approach was to prevent reactions and improve tolerance. The full dose was given on days 1–5 for all subsequent cycles to achieve better proteasome inhibition beyond the primary subunit targets (proteasome subunits Beta 5, and Beta 1 and 2 at higher doses) [16,17]. Patients with acute leukemia or non-Hodgkin lymphoma (NHL) received a single dose of intrathecal chemotherapy within 14 days of starting systemic therapy.

For cycle one, patients were pre-medicated with daily dexamethasone at 0.1 mg/kg (max 4 mg). Dexamethasone premedication was optional for subsequent cycles.

Carfilzomib doses were escalated following a rolling-6 study design until the MTD or the highest dose level was reached, whichever came first [20]. DLTs were based upon toxicities observed in cycle one only. Dose escalation was managed for each stratum independently. For stratum B, five additional patients were enrolled at the highest dose level to further assess the safety of the regimen.

Toxicities were defined according to the Common Terminology Criteria for Adverse Events (CTCAE) of the National Cancer Institute (NCI) version 4.03.

### 2.3. Disease Assessment

For stratum A, bone marrow evaluation for disease assessment was performed after every cycle. For stratum B, disease evaluation by RECIST v1.1 criteria was performed after cycles 2, 4, and 6, and then a minimum of every 3 cycles. Evaluation for lymphoma response in stratum B followed Cheson criteria. If a subject showed signs of disease progression, disease assessment was performed at that time.

Leukemia Response Assessment was defined by the following: (1) Complete remission (CR)—attainment of an M1 bone marrow (<5% blasts) with no evidence of circulating blasts or extramedullary disease and with recovery of blood counts (ANC ≥ 1000/µL and platelet count ≥ 100,000/µL). (2) Complete remission with incomplete platelet recovery (CRp)—attainment of an M1 bone marrow (<5% blasts), no evidence of circulating blasts or extramedullary disease, and with ANC ≥ 1000/µL and platelet count < 100,000/µL. (3) Complete remission with incomplete blood count recovery (CRi)—attainment of an M1 bone marrow (<5% blasts), no evidence of circulating blasts or extramedullary disease, and with ANC < 1000/µL or platelet count < 100,000/µL. (4) Partial response (PR)—a decrease of at least 50% in the percentage of blasts to 5–25% (M2 marrow) in the bone marrow aspirate. For patients with extramedullary disease only, a decrease in disease amount of at least 30%. Bone marrow must have adequate cellularity with normal hematopoietic progenitors to determine response. (5) Progressive disease (PD)—an increase of at least 25% of the absolute number of bone marrow or circulating leukemic blasts, development of new extramedullary disease or an increase in existing extramedullary disease by 20%, or other laboratory or clinical evidence of progression. (6) Relapse—morphologic relapse after CR/CRp/CRi is defined as a reappearance of leukemic blasts in the blood or ≥5% blasts in the bone marrow not attributable to any other cause (e.g., bone marrow regeneration). (7) Stable disease (SD)—patient fails to qualify for CR, CRi, CRp, PR, PD, or relapse. (8) Unevaluable (U)—aplastic or severely hypocellular marrow (<10–20% cellularity) with any blast percentage. In this instance, marrow evaluation was repeated weekly until response determination could be made.

Solid-tumor response assessment for patients with measurable disease, response, and progression were evaluated using the revised Response Evaluation Criteria in Solid Tumors (RECIST) guideline (version 1.1) [21]. For patients without measurable disease, but if disease was present on PET, MIBG, bone marrow evaluation, or serum tumor marker, response assessment was performed using that modality. Evaluation of target lesions was defined by the following: (1) Complete Response (CR)—disappearance of all target lesions. Any pathological lymph nodes (whether target or non-target) must have a reduction in short axis to <10 mm. (2) Partial Response (PR)—at least a 30% decrease in the sum of the diameters of target lesions, taking as a reference the baseline sum of diameters. (3) Progressive Disease (PD)—at least a 20% increase in the sum of the diameters of target lesions. In addition to the relative increase of 20%, the sum must also demonstrate an absolute increase of at least 5 mm (note: the appearance of one or more new lesions is also considered progression). (4) Stable Disease (SD)—neither sufficient shrinkage to qualify for PR nor sufficient increase to qualify for PD, taking as a reference the smallest sum of diameters while under study.

Evaluation of non-target lesions was defined by the following: (1) Complete Response (CR)—disappearance of all non-target lesions and normalization of tumor marker level. All lymph nodes must be non-pathological in size (<10 mm short axis). (2) NonCR/NonPD—persistence of one or more non-target lesion(s) and/or maintenance of tumor marker level above the normal limits. (3) Progressive Disease (PD)—unequivocal progression of existing non-target lesions (note: the appearance of one or more new lesions is also considered progression).

Lymphoma response assessment utilized the Cheson criteria as below [22] and was defined by the following: (1) Complete Response (CR)—disappearance of all evidence of disease. (2) Partial Response (PR)—regression of measurable disease and no new sites (>50% decrease in SPD). (3) Stable Disease (SD)—failure to attain CR/PR or PD. (4) Progressive Disease (PD)—any new lesion or increase by >50% of previously involved sites from nadir.

## 3. Results

### 3.1. Study Overview

This study was opened to enrollment on 12 February 2016 and closed on 1 March 2023. Accruals were hampered by the COVID-19 pandemic and the relocation of POETIC headquarters to Stanford in 2020. A total of 42 patients were screened, and 38 were treated (stratum A, 14; stratum B, 24) (Figure 1).

In Table 1, the median age at consent for the treatment group is shown to be 13 years (range, 3–24 years). The study participants were equally divided between males and females. Race/ethnicity for the entire group was as follows: Non-Hispanic White, 20 (52.6%); Hispanic or Latino, 13 (34.2%); Black or African American, 3 (7.9%); Asian, 1 (2.6%); and other, 1 (2.6%). Disease types are tabulated in Table 2.

### 3.2. DLTs, MTD, and Recommended Phase 2 Dose (RP2D)

For stratum A, three DLTs were observed at dose level 2: thrombocytopenia, pericarditis, and posterior reversible encephalopathy syndrome (PRES). Therefore, for patients with leukemia, the MTD and RP2D for carfilzomib given daily for 5 days, along with cyclophosphamide at 440 mg/m^2^/day and etoposide at 100 mg/m^2^/day, was 11 mg/m^2^/day (dose level 1).

Only a single DLT of PRES was observed in stratum B, at dose level 5. The MTD was not reached. With an additional dose expansion phase, a total of 11 patients were treated at dose level 5. No further toxicities meeting the DLT definition were seen. Therefore, the RP2D was determined to be dose level 5, the same cyclophosphamide and etoposide dosing as above in combination with carfilzomib at 20 mg/m^2^/day on days 1–2 and 36 mg/m^2^/day on days 3–5 for the first cycle, then 36 mg/m^2^ for days 1–5 of all subsequent cycles [16] (Table 3).

Of the 38 patients treated in this study, 21 patients received two or more cycles of therapy (stratum A, 1; stratum B, 20 [range: 2–14 cycles]).

The most common severe adverse event (SAE) was febrile neutropenia, which was observed in over 50% of patients at all dose levels. See Table 4 and Table 5 for lists of severe adverse events during cycle 1 by dose level and tumor type, respectively. AEs not categorized as SAEs during cycle 1 are presented by dose level and tumor type in Table 6 and Table 7, respectively. See the data Appendix A for additional tables for adverse events during all treatment cycles.

### 3.3. Disease Response

Disease responses were observed in both strata at all dose levels. Three objective responses were observed in stratum A: one complete remission with incomplete platelet recovery (CR_p_) (mature B-cell leukemia) and two partial responses (PRs) (AML; B ALL).

In stratum B, three PRs (germ cell tumor, non-Hodgkin lymphoma, and hepatoblastoma) and four stable diseases (SDs) (rhabdomyosarcoma, teratoma, and two cases of osteosarcoma (OS)) were observed based upon the RECIST criteria (Figure 2). Of note, the two patients with OS had no evidence of disease progression after four and five cycles, respectively.

Eight patients were enrolled in stratum B without measurable disease, primarily due to lung metastases not meeting the RECIST criteria. Of the six study patients who received >six cycles of treatment, four did not have measurable disease. One patient with synovial sarcoma and lung metastases received fourteen cycles at dose level 1, achieving pathologically confirmed CR, and is still alive and well more than seven years after completion of protocol therapy [23]. Two patients with OS received eight and nine cycles, respectively, without evidence of progressive disease when the protocol therapy was discontinued. In the first case, the patient refused to continue treatment, and in the second case, the treating physician elected to discontinue treatment to limit exposure to etoposide. One patient with hepatoblastoma received six cycles before the alpha-fetoprotein level started to rise.

## 4. Discussion

Proteasome inhibition has been shown to effectively kill tumors in cells that have acquired resistance to other anti-cancer treatments. Carfilzomib is an irreversible epoxyketone and second-generation proteasome inhibitor. Its clinical development has focused primarily on hematologic malignancies in adults, particularly multiple myeloma. In vitro studies of carfilzomib performed on solid-tumor pediatric cancer cell lines provided data and the rationale for developing a clinical trial incorporating carfilzomib into a standard chemotherapy backbone of cyclophosphamide and etoposide in patients with relapse/refractory leukemia or non-CNS solid tumors [5].

This trial used a 5-day schedule, whereas the majority of previous trials of carfilzomib used a 2-day-per-week schedule for patient convenience [15,16,18]. The 5-day dosing schedule of carfilzomib, cyclophosphamide, and etoposide, chosen for this trial to maximize synergy, was well-tolerated, with a toxicity profile similar to that observed with the chemotherapy backbone by itself as well as other multi-agent chemotherapy regimens. As expected, myelosuppression with febrile neutropenia was the main adverse event observed in the trial.

The MTD for leukemia patients (stratum A) was established at dose level 1 (carfilzomib dose 11 mg/m^2^/day × 5 days), with thrombocytopenia, PRES, and pericarditis as the observed DLTs. It is unclear whether PRES should be attributed to carfilzomib since dexamethasone, which was administered daily for 5 days prior to each carfilzomib dose to reduce infusion reactions during cycle 1, is known to cause hypertension and PRES [24]. However, since PRES is also a known toxicity of carfilzomib, we labeled it as a DLT. It was also unclear if pericarditis was due to carfilzomib as a pericardiocentesis revealed T-cell leukemia cells in the patient’s pericardial fluid, leading to our decision to enroll additional patients until a third DLT occurred.

Compared with patients enrolled in stratum B (solid tumors), patients in stratum A (leukemia) were more heavily pre-treated at study entry. There was one partial CR seen in a patient with mature B-cell leukemia. This is noteworthy because combinations with proteasome inhibitors have been used successfully to treat patients with multiple myeloma, the most differentiated mature B-cell neoplasm [18]. A response was also seen in patients with AML and pre-B ALL. The frontline COG study AAML1031 explored the use of bortezomib in AML during each chemotherapy phase, but this approach did not show improved outcomes [25]. For ALL, a different schedule of carfilzomib was studied by incorporating it into a standard four-drug ALL induction, with responses in 50% of B- and 69% of T-ALL cases [12].

The MTD was not reached for stratum B, with a single DLT of PRES observed at dose level 5. As discussed above, it was unclear if the PRES was due to the carfilzomib or the dexamethasone premedication. There were no cases of PRES reported during the second or later cycles, when patients were not required to receive dexamethasone premedication. No further toxicity concerns were evident in the dose expansion, bringing the total to 11 patients treated at dose level 5. Therefore, the RP2D for carfilzomib in combination with cyclophosphamide and etoposide was determined to be 20 mg/m^2^/day on days 1–2 and 36 mg/m^2^/day on days 3–5 for cycle 1, then 36 mg/m^2^/day for days 1–5 in all subsequent cycles [26].

There was evidence of activity in different disease groups at all dose levels. Since the trial was not designed to assess efficacy in specific disease groups, conclusions could not be drawn about the effective dosing and schedule of carfilzomib for each malignancy type. A future phase 2 trial of this regimen in specific diseases is warranted to further assess treatment efficacy.

Peripheral blood was collected for correlative studies, including the evaluation of proteosome inhibition, at multiple time points on days 1, 2, 3, and 8 of cycle 1 and days 1 and 2 of cycle 2. Samples were banked but not yet analyzed. Correlative studies looking at proteosome inhibition may help in determining the optimal dosing with this regimen. When available, correlative study data from this trial will be reported in a future paper.

Of the twenty-four patients enrolled in stratum B, fifteen (63%) had a diagnosis of sarcoma, highlighting the need for novel therapies in this patient population. One patient with synovial sarcoma (SS) and non-measurable lung disease received fourteen cycles of treatment (at dose level 1), without disease progression. This patient underwent resection of her residual lung nodules and was found to have pathological CR. The patient is alive and well more than seven years after completing protocol therapy and has since successfully conceived and given birth to a healthy child. As the outcome for metastatic SS is typically dismal, the response to this regimen is noteworthy [27].

Eleven patients in stratum B (48%) had OS. Of those patients, five had lung metastases that were technically considered non-measurable disease according to the RECIST criteria and, therefore, were not eligible for studies requiring measurable disease. RECIST has long been shown to be suboptimal for response assessment in OS [28,29,30]. Since this was a phase 1 trial, the study did not incorporate the time to progression in the assessment of response as proposed by COG for OS patients [15]. Throughout the trial, investigators observed that patients with OS significantly benefitted clinically from this combination. Incorporation of patient-reported outcome (PRO) measures may have provided additional objective insight into the impact of this treatment regimen on patients’ quality of life.

Ifosfamide in combination with etoposide, which is commonly used in patients with progressing/refractory OS, can produce significant neurotoxicity and nephrotoxicity along with low disease response rates [23]. The use of cyclophosphamide and etoposide in combination with carfilzomib may be an alternative treatment for these patients. Two out of eleven patients (18%) with OS who showed a response in this study had previously received ifosfamide in combination with etoposide or ifosfamide alone. The use of cyclophosphamide and etoposide in patients with metastatic osteosarcoma was found to have comparable results to combinations with more expensive agents but with a lesser toxicity profile [31]. The addition of carfilzomib to the cyclophosphamide/etoposide backbone could be an alternative and possibly a more effective treatment for these patients.

## 5. Conclusions

The 5-day combination of cyclophosphamide/etoposide/carfilzomib is well tolerated and easily administered in an outpatient setting, making it a viable alternative to salvage therapy for children, adolescents, and young adults with multiple malignancy types. Moving forward with an efficacy trial of this promising combination in solid tumors, particularly sarcomas, is warranted.

## Figures and Tables

**Figure 1 cancers-17-02924-f001:**
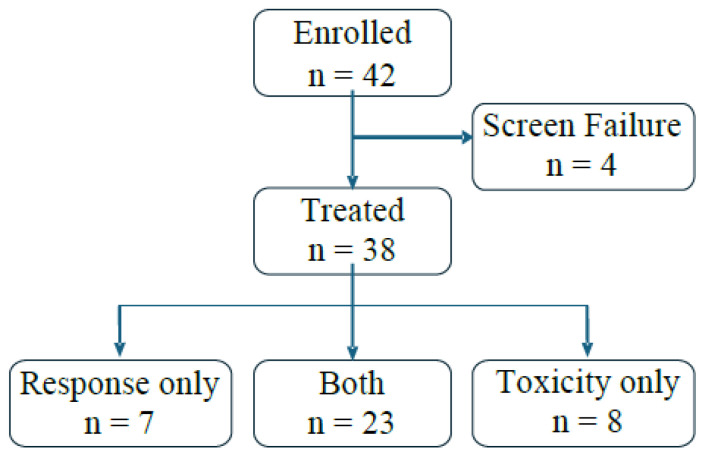
Patient flow and assessment of response and/or toxicity. The diagram summarizes patient enrollment flow through treatment to evaluability for toxicity and/or disease response.

**Figure 2 cancers-17-02924-f002:**
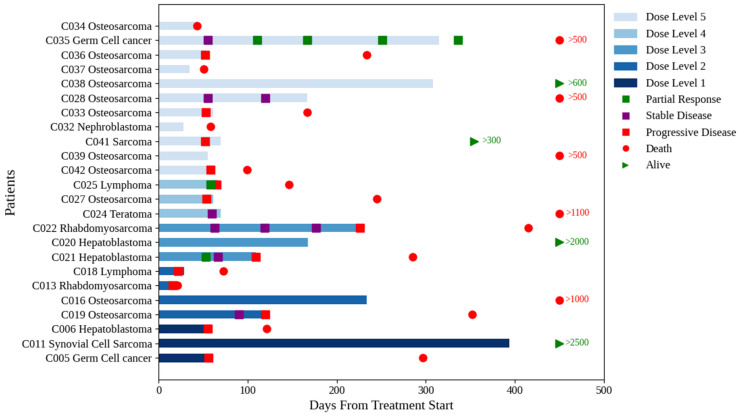
Swimmer plot for stratum B (solid tumor). The plot shows a swimmer plot for stratum B (solid tumors). Dose levels are coded by colors, from bottom level 1 to top level 5. Several patients at all dose levels transitioned to other therapies. The figure shows the number of days until death in red, and those alive in green. The numbers correspond to the last study follow-up.

**Table 1 cancers-17-02924-t001:** Patient demographic characteristics *. The table summarizes patient demographic characteristics.

	Overall (*n* = 42)	Leukemia (*n* = 16)	Solid Tumor (*n* = 26)
Age, mean (range)	13 (6–17)	7.5 (6–14)	14 (10–17)
Female, *n* (%)	21 (50)	7 (44)	14 (54)
Male, *n* (%)	21 (50)	9 (56)	12 (46)
African American, *n* (%)	4 (9.5)	1 (6)	3 (11)
Asian, *n* (%)	2 (4.8)	2 (12)	0 (0)
Hispanic, *n* (%)	15 (35.7)	6 (38)	9 (35)
Non-Hispanic White, *n* (%)	18 (42.9)	4 (25)	14 (54)
Other, *n* (%)	3 (7.1)	3 (19)	0 (0)

* Includes screen failures.

**Table 2 cancers-17-02924-t002:** Disease table. The table summarizes the number of participants enrolled by disease type and stage/histology.

Disease/Stage/Histology	*n* = 38
** Disease **	
Leukemia	14
Lymphoma	2
Solid Tumor	22
** Leukemia Type **	
Acute Bi-phenotypic Leukemia (BAL)	1
Acute Lymphoblastic Leukemia (ALL)	7
Acute Myeloid Leukemia (AML)	5
Mature B-cell Leukemia	1
** Solid Tumor Type **	
Germ Cell Tumors	3
Hepatoblastoma	3
Nephroblastoma	1
Osteosarcoma	11
Rhabdomyosarcoma	2
Sarcoma	1
Synovial Cell Sarcoma	1
** Lymphoma Type **	
Burkitt Lymphoma	1
T-cell Lymphoblastic Lymphoma	1

**Table 3 cancers-17-02924-t003:** Dose escalation levels, DLTs, MTD (stratum A), and RP2D (stratum B). The table summarizes dose-limiting toxicity for stratum A (leukemia) and B (solid tumors); dose levels 4 and 5 had split dosing of carfilzomib on days 1–2 at 20 mg/m^2^ and days 3–5 at 27 or 36 mg/m^2^.

Tumor Type	Dose Level	Carfilzomib mg/m^2^	CPM mg/m^2^	Etoposide mg/m^2^	Patients Treated	Patients Evaluable	DLTs
Leukemia	1	11 *	440	100	4	3	0
	2	15	440	100	10	5	1 pericarditis 1 thrombocytopenia 1 PRES
Solid Tumor	1	11	440	100	3	3	0
	2	15	440	100	3	2	0
	3	20	440	100	3	3	0
	4	20/27	440	100	4	4	0
	5	20/36 **	440	100	11	11	1 PRES

* MTD (stratum A). ** RP2D (stratum B). Abbreviations: CPM, cyclophosphamide; DLTs, dose-limiting toxicities; MTD, maximum tolerated dose; PRES, posterior reversible encephalopathy syndrome; RP2D, recommended phase 2 dose.

**Table 4 cancers-17-02924-t004:** Serious adverse events (SAEs) during cycle 1 by dose level in patients evaluable for toxicity (*n* = 31). Number and percentage of toxicity-evaluable subjects (*n* = 31) who experienced serious adverse events (SAEs). SAEs are categorized by dose level and reported from treatment start through the end of cycle 1. For toxicity-evaluable subjects who discontinued treatment prior to the end of cycle 1 (subjects who experienced a DLT or died before completion of the cycle), all SAEs up to the time of discontinuation are included. Values in parentheses represent the proportion of subjects (%) experiencing each listed SAE for each dose level.

Severe Adverse Event	Dose				
	11 mg/m^2^ *n* = 6	15 mg/m^2^ *n* = 8	20 mg/m^2^ *n* = 3	20/27 mg/m^2^ *n* = 3	20/36 mg/m^2^ *n* = 11
Febrile neutropenia	3 (50)	3 (37.5)	1 (33.3)	2 (66.7)	5 (45.5)
Fever	1 (16.7)	0 (0)	1 (33.3)	0 (0)	1 (9.1)
PRES	0 (0)	1 (12.5)	0 (0)	0 (0)	1 (9.1)
Anaphylaxis	0 (0)	0 (0)	0 (0)	0 (0)	1 (9.1)
Anemia	0 (0)	0 (0)	0 (0)	0 (0)	1 (9.1)
Central line infection	0 (0)	0 (0)	0 (0)	0 (0)	1 (9.1)
Creatinine increased	0 (0)	0 (0)	0 (0)	0 (0)	1 (9.1)
Epistaxis	1 (16.7)	0 (0)	0 (0)	0 (0)	0 (0)
Hypertension	0 (0)	0 (0)	0 (0)	0 (0)	1 (9.1)
Hypotension	0 (0)	0 (0)	0 (0)	0 (0)	1 (9.1)
Myalgia	0 (0)	1 (12.5)	0 (0)	0 (0)	0 (0)
Pericarditis	0 (0)	1 (12.5)	0 (0)	0 (0)	0 (0)
Renal hemorrhage	0 (0)	0 (0)	0 (0)	0 (0)	1 (9.1)
Skin infection	0 (0)	0 (0)	0 (0)	0 (0)	1 (9.1)
Vaginal infection	0 (0)	1 (12.5)	0 (0)	0 (0)	0 (0)
Wound complication	0 (0)	0 (0)	1 (33.3)	0 (0)	0 (0)

Abbreviations: PRES, posterior reversible encephalopathy syndrome; *n* in each column = number of patients at each dose level.

**Table 5 cancers-17-02924-t005:** Serious adverse events (SAEs) during cycle 1 by tumor type in patients evaluable for toxicity (*n* = 31). Number and percentage of toxicity-evaluable subjects (*n* = 31) who experienced serious adverse events (SAEs). SAEs are categorized by tumor type and reported from treatment start through the end of cycle 1. For toxicity-evaluable subjects who discontinued treatment prior to the end of cycle 1 (subjects who experienced a DLT or died before completion of the cycle), all SAEs up to the time of discontinuation are included. Values in parentheses represent the proportion of subjects (%) experiencing each listed SAE for each tumor type.

Severe Adverse Event	Tumor Type	
	Leukemia *n* = 8	Solid Tumor *n* = 23
Febrile neutropenia	5 (62.5)	9 (39.1)
Fever	1 (12.5)	2 (8.7)
PRES	1 (12.5)	1 (4.3)
Anaphylaxis	0 (0)	1 (4.3)
Anemia	0 (0)	1 (4.3)
Central line infection	0 (0)	1 (4.3)
Creatinine increased	0 (0)	1 (4.3)
Epistaxis	1 (12.5)	0 (0)
Hypertension	0 (0)	1 (4.3)
Hypotension	0 (0)	1 (4.3)
Myalgia	1 (12.5)	0 (0)
Pericarditis	1 (12.5)	0 (0)
Renal hemorrhage	0 (0)	1 (4.3)
Skin infection	0 (0)	1 (4.3)
Vaginal infection	1 (12.5)	0 (0)
Wound complication	0 (0)	1 (4.3)

Abbreviations: PRES, posterior reversible encephalopathy syndrome; *n* in each column = number of patients at each dose level.

**Table 6 cancers-17-02924-t006:** Adverse events (AEs) (grade ≥ 3) during cycle 1 by carfilzomib dose level in patients evaluable for toxicity (*n* = 31). Number and percentage of toxicity-evaluable subjects (*n* = 31) who experienced adverse events (AEs) of grade 3 or higher. AEs are categorized by dose level and reported from treatment start through the end of cycle 1. For subjects who discontinued treatment prior to the end of cycle 1 (subjects who experienced a DLT or died before completion of the cycle), all AEs up to the time of discontinuation are included. Values in parentheses represent the proportion of subjects (%) experiencing each listed AE for each dose level. Only AEs reported in more than 5% of all subjects are included.

Adverse Event	Dose				
	11 mg/m^2^ *n* = 6	15 mg/m^2^ *n* = 8	20 mg/m^2^ *n* = 3	20/27 mg/m^2^ *n* = 3	20/36 mg/m^2^ *n* = 11
Platelet count decreased	4 (66.7)	8 (100)	3 (100)	3 (100)	10 (90.9)
White blood cell decreased	5 (83.3)	7 (87.5)	3 (100)	3 (100)	10 (90.9)
Lymphocyte count decreased	4 (66.7)	6 (75)	1 (33.3)	2 (66.7)	10 (90.9)
Neutrophil count decreased	4 (66.7)	5 (62.5)	3 (100)	2 (66.7)	9 (81.8)
Anemia	4 (66.7)	5 (62.5)	3 (100)	1 (33.3)	9 (81.8)
Hypokalemia	1 (16.7)	2 (25)	1 (33.3)	0 (0)	1 (9.1)
Febrile neutropenia	1 (16.7)	3 (37.5)	0 (0)	0 (0)	0 (0)
Gamma-glutamyl transferase increased	3 (50)	1 (12.5)	0 (0)	0 (0)	0 (0)
Hypoxia	0 (0)	2 (25)	0 (0)	0 (0)	2 (18.2)
Alanine aminotransferase increased	1 (16.7)	1 (12.5)	0 (0)	1 (33.3)	0 (0)
Abdominal pain	0 (0)	1 (12.5)	0 (0)	0 (0)	1 (9.1)
Aspartate aminotransferase increased	0 (0)	1 (12.5)	0 (0)	1 (33.3)	0 (0)
Back pain	1 (16.7)	1 (12.5)	0 (0)	0 (0)	0 (0)
Hyponatremia	1 (16.7)	0 (0)	0 (0)	0 (0)	1 (9.1)
Hypotension	1 (16.7)	0 (0)	0 (0)	0 (0)	1 (9.1)
Skin infection	0 (0)	1 (12.5)	0 (0)	0 (0)	1 (9.1)

*n* in each column = number of patients at each dose level.

**Table 7 cancers-17-02924-t007:** Adverse events (AEs) (grade ≥ 3) during cycle 1 by tumor type in patients evaluable for toxicity (*n* = 31). Number and percentage of toxicity-evaluable subjects (*n* = 31) who experienced adverse events (AEs) of grade 3 or higher. AEs are categorized by tumor type and reported from treatment start through the end of cycle 1. For subjects who discontinued treatment prior to the end of cycle 1 (subjects who experienced a DLT or died before completion of the cycle), all AEs up to the time of discontinuation are included. Values in parentheses represent the proportion of subjects (%) experiencing each listed AE for each tumor type. Only AEs reported in more than 5% of all subjects are included.

Adverse Event	Tumor Type	
	Leukemia *n* = 8	Solid Tumor *n* = 23
Platelet count decreased	8 (100)	20 (87)
White blood cell decreased	8 (100)	20 (87)
Lymphocyte count decreased	8 (100)	15 (65.2)
Neutrophil count decreased	6 (75)	17 (73.9)
Anemia	8 (100)	14 (60.9)
Hypokalemia	3 (37.5)	2 (8.7)
Febrile neutropenia	4 (50)	0 (0)
Gamma-glutamyl transferase increased	3 (37.5)	1 (4.3)
Hypoxia	2 (25)	2 (8.7)
Alanine aminotransferase increased	1 (12.5)	2 (8.7)
Abdominal pain	1 (12.5)	1 (4.3)
Aspartate aminotransferase increased	1 (12.5)	1 (4.3)
Back pain	2 (25)	0 (0)
Hyponatremia	1 (12.5)	1 (4.3)
Hypotension	1 (12.5)	1 (4.3)
Skin infection	1 (12.5)	1 (4.3)

*n* in each column = number of patients at each dose level.

## Data Availability

Research data supporting this publication are available from the POETIC consortium upon email request to poeticrdmc@stanford.edu.

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
