# Peer review of "A Phase I Study of Carfilzomib with Cyclophosphamide and Etoposide in Relapsed and Refractory Leukemia and Solid Tumors"

_cancers, 2025, doi:10.3390/cancers17172924_

Round 1

Reviewer 1 Report

Comments and Suggestions for Authors

This study tested the use of carfilzomib, a second generation proteasome inhibitor, in children and young adults, many with osteosarcoma. As I understand, the aim was to rescue the patients with a new treatment to see if they could be "rescued" from very bad prognosis disease. Phase I studies carry the most potential risk, but phase I study may help some patients.
Text is well written, it is easy to read and understand.

Comments:

(1) Could you please show the chemical structure of cafilzomib?

(2) This drug is a proteasome inhibitor. Could you please describe the proteasome in the introduction?  
*For example, "The proteasome is a protein-destroying apparatus involved in many essential cellular functions, such as regulation of cell cycle, cell differentiation, signal transduction pathways, antigen processing for appropriate immune responses, stress signaling, inflammatory responses, and apoptosis."
"The proteasome is a central component of the protein degradation machinery in eukaryotic cells"
"Both transformed and normal cells depend on the function of the proteasome to control the expression of proteins linked to cell survival and proliferation"

**What is the structure of proteasome?
*** What is the mechanism of action of the proteasome inhibitors and cafilzomib? Does it affect the 19S regulatory particle or the 20S core particle?

(3) Lines 63-75. For each statement from 1 to 6, I think an reference should be added, doesn't it?

(4) Line 114-115. "acute leukemia or non-Hodgkin lymphoma (NHL)". In Table 1, only leukemia (n=16) and solid tumor (n=26) is mentioned. Are NHL included in leukemia group?

(5) Lines 147-150. Could you please explain why the stratification according to ethinic groups is necessary? Is the metabolism of proteasome different?

(6) In methods, the type of leukemia or solid tumors could be described with more detail. It appears that most solid tumors are osteosarcomas, is this correct?

(7) Looking at Table 4. It seems many patients developed serious adverse events, doesn't it?

(8) Line 226. Sorry to ask but, why was the patient with teratoma included in the study? Shoudl not the tumor removed surgically?

(9) What are the criteria of PR, SD, PD, etc.? In case of NHL you may use Cheson (and consecuently updated) criteria, but for solid tumors?

(10) Regarding Figure 2. As I understand, this study lacks control patients, is this right?  (cyclophosphamide + etoposide without carfilzomib)

(11) In Figure 2 several patients died within the first 3 or 6 months. Whas the dead related to the disease (neoplasia), or could be atribuited to a side-effect?

(12) Regarding "Several patients at all dose levels transitioned to other therapies". What is the cause of changing therapy?

(13) Why were immune checkpoint inhibitors not used in addition to proteasome inhibitor?

(14) Some years ago, it was said that proteasome inhibitors did not work well in solid tumors. Is this right?

(15) Line 322. It is stated to 5-day combination is well tolerated. However, SAEs were identified as well, is this right?

(16) Regarding conclusions of abstract. "A 5-day schedule of carfilzomib/cyclophosphamide/etoposide was well tol- 47
erated in patients with solid tumors." Please show what lines of results paragraph describe this output.

(17) Regarding abstract conclusion "Patients with sarcomas benefited most" Please highlight lines with this result. (just to double check)

(18) I understand that phase 0 clinical trial was completed previously?

(19) The text has many details. Please confirm that there are not mistakes in type of treatment, dose, results, etc.

(20) Should raw data we included as supplementary in form of tabulated data?

(21) Is this study registered in clinical trial website or has NIH approvals?

Author Response

General Comment:

This study tested the use of carfilzomib, a second-generation proteasome inhibitor, in children and young adults, many with osteosarcoma. As I understand, the aim was to rescue the patients with a new treatment to see if they could be "rescued" from very bad prognosis disease. Phase I studies carry the most potential risk, but phase I study may help some patients.
Text is well written, it is easy to read and understand.

Comments (1) Could you please show the chemical structure of cafilzomib?

Response (1) We could, but it may not add information for readers, we prefer your suggestion (2) below as it provides information on the drug mechanism of action for all readers. We usually add it in a clinical trial report when it is a new in class agent. We acknowledge the suggestion, we will add the following reference.

  1. Jayaweera SPE, Wanigasinghe Kanakanamge SP, Rajalingam D, Silva GN. Carfilzomib: A Promising Proteasome Inhibitor for the Treatment of Relapsed and Refractory Multiple Myeloma. Front Oncol. 2021 Nov 10;11:740796.

Comments (2) This drug is a proteasome inhibitor. Could you please describe the proteasome in the introduction?

Response (2) Thank you, the description below was added to the article as you suggested, as it conveys more information than the chemical structure to most readers.

For example, "The proteasome is a protein-destroying apparatus involved in many essential cellular functions, such as regulation of cell cycle, cell differentiation, signal transduction pathways, antigen processing for appropriate immune responses, stress signaling, inflammatory responses, and apoptosis. The proteasome is a central component of the protein degradation machinery in eukaryotic cells. Both transformed and normal cells depend on the function of the proteasome to control the expression of proteins linked to cell survival and proliferation."

What is the structure of proteasome?

Response (2) We usually add it in a clinical trial report when it is a new in class target.

 What is the mechanism of action of the proteasome inhibitors and cafilzomib?

Response (2) It is an irreversible inhibitor of the proteasome and inhibits its chymotrypsin-like activity

Does it affect the 19S regulatory particle or the 20S core particle?

Response (2) It affects the chymotrypsin-like activity of the 20S core, it does not affect the 19S regulatory particle.

 References:

  1. KYPROLIS. FDA Drug Label. Food and Drug Administration. Updated date: 2025-06-18
  2. Carfilzomib Can Induce Tumor Cell Death Through Selective Inhibition of the Chymotrypsin-Like Activity of the Proteasome. Parlati F, Lee SJ, Aujay M, et al. Blood. 2009;114(16):3439-47. doi:10.1182/blood-2009-05-223677.
  3. Crystal Structure of the Human 20S Proteasome in Complex With Carfilzomib. Harshbarger W, Miller C, Diedrich C, Sacchettini J. Structure (London, England: 1993). 2015;23(2):418-24. doi:10.1016/j.str.2014.11.017

Comments (3) Lines 63-75. For each statement from 1 to 6, I think a reference should be added, doesn't it?

Response (3) We reviewed publications and added references shown below to the manuscript:

1.Fractionated Cyclophosphamide and Etoposide for Children With Advanced or Refractory Solid Tumors: A Phase II Window Study. Mantadakis E, Herrera L, Leavey PJ, et al.Journal of Clinical Oncology: Official Journal of the American Society of Clinical Oncology. 2000;18(13):2576-81. doi:10.1200/JCO.2000.18.13.2576.

2.A Dose-Intensive, Cyclophosphamide-Based Regimen for the Treatment of Recurrent/­Progressive or Advanced Solid Tumors of Childhood: A Report From the Australia and New Zealand Children's Cancer Study Group. Carpenter PA, White L, McCowage GB, et al. Cancer. 1997;80(3):489-96. doi:10.1002/(sici)1097-0142(19970801)80:3<489:aid-cncr17>3.0.co;2-t.

3.Clofarabine, Cyclophosphamide and Etoposide for the Treatment of Relapsed or Resistant Acute Leukemia in Pediatric Patients. Miano M, Pistorio A, Putti MC, et al. Leukemia & Lymphoma. 2012;53(9):1693-8. doi:10.3109/10428194.2012.663915.

Comments (4) Line 114-115. "Acute leukemia or non-Hodgkin lymphoma (NHL)". In Table 1, only leukemia (n=16) and solid tumor (n=26) is mentioned. Are NHL included in leukemia group?

Response (4) The single NHL case was included in the solid tumor group as the response assessments are like those for solid tumors. Lymphomas are frequently grouped with solid tumors in pediatric oncology phase I studies. The inclusion of lymphoma into the solid tumor group is mentioned in the eligibility criteria section.

Comments (5) Lines 147-150. Could you please explain why the stratification according to ethnic groups is necessary? Is the metabolism of proteasome different?

Response (5) This is done for all pediatric studies as occasionally ethnicity or race may be associated with pharmacogenomic features and it may be relevant if the drug is widely used. We do not have information that proteasome metabolism is different across ethnic or racial groups, but there are ethnic specific genetic variants in the proteasome, but of no known clinical significance.

  1. Proteasome Beta Subunit Pharmacogenomics: Gene Resequencing and Functional Genomics. Wang L, Kumar S, Fridley BL, et al. Clinical Cancer Research: An Official Journal of the American Association for Cancer Research. 2008;14(11):3503-13. doi:10.1158/1078-0432.CCR-07-5150.

Comments (6) In methods, the type of leukemia or solid tumors could be described with more detail. It appears that most solid tumors are osteosarcomas, is this correct?

Response (6) We had the description of leukemia cases in supplement "POE14-01 Publication Supplement_26Jun25" Table 6. We added this table to the manuscript, and the supplement has been edited and renamed to "POE14-01 Publication Supplement_29Aug25"

Of 24 solid tumors, 11 were osteosarcomas.

Comments (7) Looking at Table 4. It seems many patients developed serious adverse events, doesn't it?

Response (7) There were no Suspected Unexpected Serious Adverse Reaction (SUSARS), but given the backbone we used, many of the AEs and SAEs are common adverse events seen in pediatric cancer treatments with or without carfilzomib. Specifically, febrile neutropenia. We did not find new treatment emergent adverse events (TEAEs) specifically related to carfilzomib or this combination and refer to this point in the discussion lines (191-200, and 260-263).

Comments (8) Line 226. Sorry to ask but, why was the patient with teratoma included in the study? Should not the tumor removed surgically?

Response (8) You are correct, newly diagnosed teratomas (benign or malignant) are amenable to surgical resection. In this case the patient suffered from a recurrent stage 3 teratoma, not be amenable to resection. The malignant teratoma in the lung failed four prior treatments over a span of four years, and was enrolled on our study for a fifth attempt to control disease.

Comments (9) What are the criteria of PR, SD, PD, etc.? In case of NHL you may use Cheson (and consecuently updated) criteria, but for solid tumors?

Response (9) We added specific criteria for leukemia and solid tumors (including NHL). New references 21 and 28 in the manuscript.

Comments (10) Regarding Figure 2. As I understand, this study lacks control patients, is this right?  (cyclophosphamide + etoposide without carfilzomib).

Response (10) Correct, usually we do not use controls in pediatric early phase studies as there are few patients, and the goal is to introduce the use of new agents on an active treatment backbone previously used in the pediatric cancers under study or in adult cancers.

Comments (11) In Figure 2 several patients died within the first 3 or 6 months. Whas the dead related to the disease (neoplasia), or could be attributed to a side-effect?

Response (11) Thirteen patients died due to disease progression, and one additional patient, C034 died of sepsis 11-days after last treatment. In frontline cancer treatments of children, we observe toxic deaths at a rate of 0.1-1%.

Comments (12) Regarding "Several patients at all dose levels transitioned to other therapies". What is the cause of changing therapy?

Response (12) Disease progression, or treating physician choice, usually empirically added oral targeted therapy agents for better quality of life.

Comments (13) Why were immune checkpoint inhibitors not used in addition to proteasome inhibitor?

Response (13) This was not included in study design, during study progress the first immune checkpoint inhibitor studies started in pediatrics. Our consortium participated in the study of Atezolizumab.

1.Atezolizumab for Children and Young Adults With Previously Treated Solid Tumours, Non-Hodgkin Lymphoma, and Hodgkin Lymphoma (iMATRIX): A Multicentre Phase 1-2 Study.Geoerger B, Zwaan CM, Marshall LV, et al. The Lancet. Oncology. 2020;21(1):134-144.  doi:10.1016/S1470-2045(19)30693-X.

Comments (14) Some years ago, it was said that proteasome inhibitors did not work well in solid tumors. Is this right?

Response (14) There is evidence of activity in adult and childhood cancers. See references below.

  1. Exploring the Role and Clinical Implications of Proteasome Inhibition in Medulloblastoma. Hoerig CM, Plant-Fox AS, Pulley MD, Di K, Bota DA.1Pediatric Blood & Cancer. 2021;68(10):e29168. doi:10.1002/pbc.29168.
  2. Proteasome Inhibition to Maximize the Apoptotic Potential of Cytokine Therapy for Murine Neuroblastoma Tumors. Khan T, Stauffer JK, Williams R, et al. Journal of Immunology (Baltimore, Md.: 1950). 2006;176(10):6302-12. doi:10.4049/jimmunol.176.10.6302.
  3. Phase I Study of the Proteasome Inhibitor Bortezomib in Pediatric Patients With Refractory Solid Tumors: A Children's Oncology Group Study (ADVL0015). Blaney SM, Bernstein M, Neville K, et al. Journal of Clinical Oncology: Official Journal of the American Society of Clinical Oncology. 2004;22(23):4804-9. doi:10.1200/JCO.2004.12.185.
  4. The Ubiquitin-Proteasome Pathway in Adult and Pediatric Brain Tumors: Biological Insights and Therapeutic Opportunities. Zaky W, Manton C, Miller CP, et al. Cancer Metastasis Reviews. 2017;36(4):617-633. doi:10.1007/s10555-017-9700-2.

Comments (15) Line 322. It is stated to 5-day combination is well tolerated. However, SAEs were identified as well, is this right?

Response (15) SAEs/AEs are within what is expected in relapse/refractory sarcomas and leukemia on non-experimental frontline salvage treatments. Given the backbone we used, many of the AEs and SAEs are like the adverse events described in pediatric cancer treatments with most used chemotherapy treatments with or without carfilzomib. Specifically, febrile neutropenia. We did not find new treatment emergent adverse events (TEAEs) specifically related to carfilzomib or this combination and refer to this point in the discussion lines (191-200, and 260-263).

Comments (16) Regarding conclusions of abstract. "A 5-day schedule of carfilzomib/cyclophosphamide/etoposide was well tolerated in patients with solid tumors." Please show what lines of results paragraph describe this output.

Response (16) There were no SUSARS, but given the backbone we used, many of the AEs and SAEs are similar to adverse events, specifically febrile neutropenia, described in pediatric cancer treatments with most used chemotherapy treatments with or without carfilzomib. We did not find new treatment emergent adverse events (TEAEs) specifically related to carfilzomib or this combination and refer to this point in the discussion lines 260-263.

Comments (17) Regarding abstract conclusion "Patients with sarcomas benefited most" Please highlight lines with this result. (just to double check).

Response (17) The information is noted in lines 231-241, and in the discussion on lines 299-316 of the reviewed manuscript. Upon our major revision, the numbers in the text will be different.

Comments (18) I understand that phase 0 clinical trial was completed previously?

Response (18) No phase 0 studies were completed, but preclinical testing of leukemia and solid tumor cell lines were tested with the study drug combination. References were cited in the paper:

  1. Thakur S, Ruan Y, Jayanthan A, Boklan J, Narendran A. Cytotoxicity and Target Modulation in Pediatric Solid Tumors by the Proteasome Inhibitor Carfilzomib. Curr Cancer Drug Targets. Published online May 3, 2021. doi:10.2174/1568009621666210504085527; already cited.
  2. Swift L, Jayanthan A, Ruan Y, et al. Targeting the Proteasome in Refractory Pediatric Leukemia Cells: Characterization of Effective Cytotoxicity of Carfilzomib. Target Oncol. 2018;13(6):779-793. doi:10.1007/s11523-018-0603-0. Was added to references.

Comments (19) The text has many details. Please confirm that there are not mistakes in type of treatment, dose, results, etc.

Response (19) Thank you, we reviewed the manuscript and our electronic capture database carefully and did not find any mistakes.

Comments (20) Should raw data we included as supplementary in form of tabulated data?  

Response (20) Please see "POE14-01 Publication Supplement_29Aug25" with raw data tabulated.

Comments (21) Is this study registered in clinical trial website or has NIH approvals?

Response (21) We included in the manuscript: Clinical Trial Information: ClinicalTrials.gov Identifier: NCT02512926, but the journal removed it.

Reviewer 2 Report

Comments and Suggestions for Authors

The manuscript "A Phase I Study of Carfilzomib with Cyclophosphamide and 2 Etoposide in Relapsed and Refractory Leukemia and Solid Tumors " by Boklan J et al is an interesting and important report on outcome of Carfilzomib combination therapy in relapse/refractory leukemia and solid tumor as a phase I trial study. As Carfilzomib has shown efficacy in adults, it is important to  investigate its treatment outcome in pediatric and young adult population. 

The study is well written and presented with adequate data on the results of the trial. A few modification would increase the readers understanding of the trial results.

In the methodology section, disease assessment description is written briefly and could be described more about the different parameters on which the assessment was done.

In the correlative study, the authors have mentioned that the samples were biobanked but the analysis was not performed. If the results are not to be shown in the current report, it can be removed from the methodology and just mention the same in the discussion part as done at line #295-296.

The result and discussion part is well described and doesn't need modification.

Author Response

Comment 1: n the methodology section, disease assessment description is written briefly and could be described more about the different parameters on which the assessment was done.

Response 1:The parameters for assessments were added to the manuscript.

Comment 2:In the correlative study, the authors have mentioned that the samples were biobanked but the analysis was not performed. If the results are not to be shown in the current report, it can be removed from the methodology and just mention the same in the discussion part as done at line #295-296.

Response 2:We amended the manuscript to accept your suggestion.